# Proteomic Analysis of Rat Duodenum Reveals the Modulatory Effect of Boron Supplementation on Immune Activity

**DOI:** 10.3390/genes14081560

**Published:** 2023-07-30

**Authors:** Chunfang Zhao, Shuqin Chen, Yujiao Han, Feng Zhang, Man Ren, Qianqian Hu, Pengfei Ye, Xiaojin Li, Erhui Jin, Shenghe Li

**Affiliations:** 1College of Animal Science, Anhui Science and Technology University, No. 9 Donghua Road, Fengyang County, Chuzhou 233100, China; zhaochf@ahstu.edu.cn (C.Z.); chenshq@ahstu.edu.cn (S.C.); hanyj@ahstu.edu.cn (Y.H.); zhangfeng@ahstu.edu.cn (F.Z.); renm@ahstu.edu.cn (M.R.); huqq@ahstu.edu.cn (Q.H.); yepf@ahstu.edu.cn (P.Y.); lixj@ahstu.edu.cn (X.L.); lish@ahstu.edu.cn (S.L.); 2Anhui Province Key Laboratory of Animal Nutritional Regulation and Health, No. 9 Donghua Road, Fengyang County, Chuzhou 233100, China

**Keywords:** rat, boron, proteome, immune activity, biological signal pathway

## Abstract

The proper supplementation of boron, an essential trace element, can enhance animal immune function. We utilized the method of TMT peptide labeling in conjunction with LC-MS/MS quantitative proteomics for the purpose of examining the effects of boric acid on a rat model and analyzing proteins from the duodenum. In total, 5594 proteins were obtained from the 0, 10, and 320 mg/L boron treatment groups. Two hundred eighty-four proteins that exhibit differential expression were detected. Among the comparison, groups of 0 vs. 10 mg/L, 0 vs. 320 mg/L, and 10 vs. 320 mg/L of boron, 110, 32, and 179 proteins, respectively, demonstrated differential expression. The results revealed that these differential expression proteins (DEPs) mainly clustered into two profiles. GO annotations suggested that most of the DEPs played a role in the immune system process, in which 2′-5′-oligoadenylate synthetase-like, myxovirus resistance 1, myxovirus resistance 2, dynein cytoplasmic 1 intermediate chain 1, and coiled-coil domain containing 88B showed differential expression. The DEPs had demonstrated an augmentation in the signaling pathways, which primarily include phagosome, antigen processing, and presentation, as well as cell adhesion molecules (CAMs). Our study found that immune responses in the duodenum were enhanced by lower doses of boron and that this effect is likely mediated by changes in protein expression patterns in related signaling pathways. It offers an in-depth understanding of the underlying molecular mechanisms that lead to immune modulation in rats subjected to dietary boron treatment.

## 1. Introduction

Boron has been recently classified as a prebiotic micronutrient, being defined as essential in host-bacterial symbiosis in human health, and is predominantly sourced from many kinds of food and drinking water in the human diet [1]. Additionally, boron has widespread applications in various fields, such as medical science, manufacturing, farming, and beauty products [2,3]. Boron displays pleiotropic effects and serves multiple functions in the human body, including but not limited to physiological processes of bone formation, neurological well-being, cellular membrane stability, mineral balance, immunological regulation, and hormonal equilibrium [4,5,6,7,8].

Akbari et al. (2022) evaluated the potential therapeutic benefits of boron citrate and oleoylethanolamide addition for the treatment of COVID-19 patients [9]. The study found that the utilization of boron citrate (5 mg twice a day) in isolation or in conjunction with oleoylethanolamide (200 mg twice a day) has the potential to ameliorate certain clinical and biochemical measures in individuals afflicted with COVID-19 [9]. Bouchareb et al. (2020) showed that the introduction of boron (4 mg/kg) had a beneficial effect on heart failure induced by myocardial infarction and reduced the severity of cardiac fibrosis and apoptosis, two primary characteristics of heart failure [10]. Importantly, boron is a crucial element in the potential induction of the cell cycle entry of cardiomyocytes, which could lead to the regeneration of cardiac tissue in the aftermath of injury. Mohammed et al. (2023) found that boron derivatives sodium pentaborate pentahydrate (500 μg/mL) and sodium perborate tetrahydrate (50 μg/mL) suppressed proliferation and induced apoptosis in breast cancer cell lines by means of suppressing the monopolar spindle-one-binder (MOB1) protein [11]. However, the upregulation of programmed death-ligand 1 (PD-L1) protein and downregulation of pro-inflammatory cytokines were observed, leading to the suppression of effector T cells that were specifically activated against breast cancer cells. The follicular fluid was discovered to contain a considerably greater amount of boron than the serum. Additionally, a global proteomics analysis revealed a favorable connection between the concentration of boron and the levels of specific immunoglobulins [12]. Wang et al. (2021) report that exposure to boron (100 mg/kg) in male rats via oral gavage led to reproductive toxicity and metabolic perturbations. Testicular hormones were upregulated, and there was a disruption in lipid and amino acid metabolism, which are linked with the process of inflammation, the protection of antioxidation, the utilization of energy, the synthesis of steroid hormones, and the irregularities of lipids [13].

Several studies have recently concentrated on investigating the potential impact of the dietary supplement boron on the activities of the immune system in various animal species, such as livestock. The effect of boron on biology is dependent on the amount given and displays a U-shaped response, requiring verification of the appropriate range [14]. Cho et al. (2022) found that the introduction of a moderate amount of boron (5 mg/kg) in the diet of weaned pigs led to an enhancement in their growth performance, crude protein digestibility, and reduction in the diarrhea index [15]. Sizmaz et al. (2017) found that the administration of boron (0, 200, 300, and 400 mg/kg) through dietary supplementation had a varying impact on the microbial fermentation of the rumen and the abundance of protozoa [16]. It was observed that the influence of boron was dependent on the dosage. However, the concentration of boron in the rumen fluid did not witness a simultaneous increase with the rise in dosage in the diet of yearling rams. Sharma et al. (2020) reported that the addition of boron at a concentration of 200 ppm in the diet of peripartum Murrah buffaloes was found to promote metabolic adaptations that enhance calcium homeostasis, bone metabolism, and antioxidant status [17]. However, the study did not observe significant additional benefits at higher levels of boron supplementation. Boron administered in drinking water has been shown to modulate the immune reactions in ostrich chicks by influencing the calcium and MAPK signaling pathways within the thymus [18].

The immune system in the intestine is tasked with performing the vital function of upholding a delicate equilibrium between defense, tolerance, and tissue repair, which is regarded as the largest compartment of the immune system, and the optimal functioning of T-cells and their ability to produce cytokines are integral for the physiological development and maturity of the intestine [19,20]. The duodenum is of utmost importance as the initial part of the small intestine. Its anatomical position after the stomach allows it to receive partially digested food, bile, and pancreatic enzymes. It plays a vital role in neutralizing stomach acid, breaking down nutrients, and aiding in nutrient absorption. Additionally, the duodenum houses immune cells as part of the gut-associated lymphoid tissue, contributing to the immunological defense and promoting tolerance to harmless substances [21]. This small but crucial segment is essential for digestion, nutrient assimilation, and maintaining a healthy immune response in the body [22]. The study conducted by Abtisam et al. (2023) found that the addition of boron (15, 25, and 45 mg/L) and nanoboron (25 and 45 mg/L) to chickens’ drinking water can enhance the structure of the small intestine, especially at high nanoboron concentrations [23]. Supplementation of boron in drinking water at 40 and 80 mg/L levels could enhance the structure and function of the duodenum in rats, while higher doses of 320–640 mg/L could have a significant inhibitory effect, according to Liu et al.’s (2021) study investigating the effects of different doses of boron on the microstructure of the duodenum in rats [24]. The study conducted by Zhao et al. (2021) determined that boron (10 mg/L) supplementation with distilled drinking water in rats can enhance the immune response in the duodenum, specifically through the activation of signaling pathways related to influenza A, herpes simplex infection, cytosolic DNA-sensing, and antigen processing and presentation, with low doses of boron being more effective [25].

In our preceding research endeavors, we have analyzed the impact of boron on the microstructure of the duodenum, expression of secretory immunoglobulin A and tight junction protein, cell proliferation, and apoptosis in rats [24]. Furthermore, we have identified some differentially expressed genes (DEGs) by means of RNA-Seq. In order to carry out a conjoint analysis involving transcriptome data and to further authenticate DEGs that played a critical role in biological signaling pathways closely linked to boron administration, we have undertaken proteome analysis to screen for DEPs at varying doses of boron supplementation in rats. The objective of this research was to scrutinize the influence of boron on protein expression in rat intestines by scrutinizing the protein profile of the duodenum via TMT peptide labeling and LC-MS/MS quantitative proteomics following the administration of different doses of boron.

Boron is authorized for use in food and dietary supplements with concentration limits set by regulatory agencies. Boron plays various roles in maintaining bone health, arthritis and joint health, heart health, hormone regulation, wound healing, diabetes management, and cancer treatment, and is used in contraceptives, oral hygiene products, and vaccines [26]. Boron compounds have been found to have antimicrobial properties, which could be useful in developing new treatments for infections, and some research suggests they could be used in targeted cancer therapy, particularly in boron neutron capture therapy [26].

## 2. Methods

### 2.1. Experimental Animals and Design

Thirty-six male Sprague Dawley rats at 23 days of age, all of which were specific pathogen-free (SPF), with a weight of 53 ± 2 g and lot No.: SCXK Zhe 2014-001, were acquired from the Qinglongshan experimental animal breeding farm located in Jiangning District, Nanjing, China [25]. The use of these experimental animals (protocol number: AHSTULL2017018) was subjected to review and approval by the Anhui Laboratory Animal Care Committee. All animal experimental procedures were strictly carried out in accordance with the guidelines for Laboratory Animal Care and Use and also complied with the standards of the Guide for National Laboratory Animal Healthcare and Use. The housing, handling, and disposal of rats were in accordance with our previously published article [25].

The allocation of the rats was carried out in a randomized manner, with three groups being formed, each consisting of 12 rats. The first group served as the control and was given distilled water, while the other two groups were administered drinking water that had been enriched with varied concentrations of boron. Boric acid, having a degree of purity greater than 99.5%, and a boron concentration surpassing 17.4 g per 100 g, was employed as a boron source. The additional dose of boron was consistent with our previous study [24,25]. The drinking water was infused with boron concentrations of 10 mg/L and 320 mg/L, representing boric acid quantities of 57.21 mg/L and 1830.66 mg/L, respectively. The rats were housed separately in well-ventilated enclosures that were maintained at a temperature range of 24–28 °C and a humidity range of 50–65%. A 12 h light and 12 h dark cycle was implemented, and the animals were allowed unrestricted access to food and water. Food and water intake were recorded, and the feed for the rats was bought from the Qinglongshan breeding farm. The fundamental nutritional components of the feed consisted of crude protein of no less than 18.15%, crude fat of no less than 4.03%, crude fiber of no less than 5.12%, crude ash of no less than 7.94%, calcium content of 1.43%, phosphorus content of 0.87%, and boron content of 1.96 milligrams per kilogram.

### 2.2. Sample Collection and Preparation

An anesthetic breathing system (ABS) was used to anesthetize three rats from each group. After 60 days of experimentation, anesthesia was induced using 4–5% isoflurane, followed by 1–2% for maintenance. The rats were euthanized by puncturing the heart, and the duodenum was promptly collected. It was then ground using liquid nitrogen.

### 2.3. Protein Extraction and Quantification

A frozen sample was taken in an appropriate amount and treated with protein lysis buffer containing 8 M urea and 1% SDS and a protease inhibitor cocktail in a 1:12 ratio. The mixture was then subjected to high-speed shaking in a tissue disrupter for 40 s, followed by on-ice lysis for 30 min. Following the lysis process, the resultant mixture was subjected to a chilling centrifugation at 4 °C and 16,000× *g* for 30 min, which led to the separation of the supernatant that housed the protein of interest. The protein concentration was quantified using a commercial BCA agent (Beyotime Co., Ltd., Beijing, China) in 96-well plates.

### 2.4. Trypsin Digestion and TMT Peptide Labeling

A total of 100 µg of protein was combined with lysis buffer to a total volume of 100 µL. Subsequently, the sample was supplemented with a solution containing 10 mM TCEP and incubated at a temperature of 37 °C for a duration of 60 min. Following this, a solution containing 40 mM iodoacetamide was added to the mixture, which was then kept away from light at room temperature for 40 min. Further, each tube was supplemented with acetone that was pre-cooled, with a ratio of 6:1 of acetone to sample volume. The sample was then subjected to precipitation at a temperature of −20 °C for 4 h and centrifuged at a speed of 10,000× *g* for 20 min. The resultant precipitate was collected and dissolved completely in 100 µL of 100 mM TEAB, following which trypsin was added at a mass ratio of 1:50 (enzyme to protein). The sample was then subjected to digestion overnight at a temperature of 37 °C.

The TMT reagent (lot No.: 90111, Thermo, Rockford, IL, USA) was extracted from a temperature of −20 °C and permitted to achieve equilibrium with the surrounding environment. Acetonitrile was subsequently introduced, followed by thorough mixing and centrifugation. For each 100 mg of peptide, a single tube of TMT reagent was introduced and subjected to an incubation period of 2 h at ambient temperature [27]. Following this, hydroxylamine was included and underwent an incubation period of 15 min at the same temperature. Three sets of three biological replicates each were designated for the control, 10 mg/L and 320 mg/L boron groups, respectively. The control group replicates were distinguished by TMT10-126, TMT10-127N, and TMT10-127C labels, while the 10 mg/L boron group replicates were identified by TMT10-128N, TMT10-129N, and TMT10-129C labels. The 320 mg/L boron group replicates were subjected to labeling with TMT10-130N, TMT10-130C, and TMT10-131. The equimolar products were combined and subjected to vacuum concentration for drying.

### 2.5. Reversed-Phase High-Performance Liquid Chromatography (HPLC) Analysis

To isolate the marked peptides, an equivalent amount of 100 μg of each sample from the three distinct groups was combined. The resulting peptide combination was subjected to segregation through reversed-phase chromatography, utilizing an ACQUITY UPLC instrument (Waters, Milford, MA, USA), which featured an ACQUITY UPLC BEH C18 analytical column (1.7 μm, 2.1 mm × 150 mm, Waters, Milford, MA, USA), as previously described [28]. Prior to being loaded, the sample underwent dissolution in solvent A, which was composed of 2% acetonitrile and 10 mmol/L ammonium formate (pH 10.0), whilst solvent B consisted of 80% acetonitrile and 10 mmol/L ammonium formate (pH 10.0). The experiment utilized the following gradient conditions: an initial 0–3.8% concentration of solvent B for a duration of 15 min, followed by a 3.8–24% concentration of solvent B for a duration of 18 min, then a 24–100% concentration of solvent B for a duration of 5 min, and finally a 100–0% concentration of solvent B for a duration of 6 min. Peptides underwent separation through a 66 min gradient protocol, whereby a flow rate of 0.2 mL/min was employed at a wavelength of 214 nm. A total of 20 fractions were collected and merged into 10 fractions. After undergoing vacuum centrifugal concentration, the precipitate was analyzed through the employment of mass spectrometry. 

### 2.6. LC-MS/MS Analysis

Nano-LC-MS/MS investigations were conducted using a Thermo Xcalibur 4.0 system coupled with a Q Exactive (Thermo) mass spectrometer, as previously described [29]. Each specimen of the peptide was isolated through the utilization of an analytical column (75 μm × 25 cm, Thermo). The mobile phase for peptide fraction analysis included mobile phase A (2% acetonitrile with 0.1% formic acid) and mobile phase B (80% acetonitrile with 0.1% formic acid). The LC-MS/MS analysis was executed with a specific set of gradient conditions, which consisted of a solvent B range of 0% to 23% over a duration of 63 min, followed by an increase from 23% to 100% solvent B within 32 min, and lastly, a decrease from 100% to 0% solvent B over a period of 25 min. The analytical column was operated at a flow rate of 300 nL/min.

The mass spectra were obtained in the positive-ion mode using the DDA mode. Peptide precursor MS scans ranging from 350 to 1300 *m*/*z* were conducted in the orbitrap at a resolution of 35 K. Tandem MS was conducted through the isolation of the precursor ion, fragmentation by HCD with a normalized collision energy of 30, and rapid scanning of the MS analysis in the ion trap at 1, 5 Th quadrupole. The precursor ion was disregarded for additional fragmentation following two MS/MS analyses in 18 s.

### 2.7. Proteins Identification and Quantification

The Rattus Norvegicus database was analyzed and quantified utilizing the Proteome Discoverer 2.1 software. The allowed deviation for precursor and fragment ions was 10 ppm and 0.02 Da, respectively. Trypsin digestion was allowed with 2 missed cleavages, and dynamic modifications were selected for protein acetylation at the N-terminus and methionine oxidation. Static modifications were chosen for the N-terminal TMT6-plex label, lysine TMT6-plex label, and cysteine carbamidomethylation. Proteins that exhibited a *p*-value threshold < 0.05 were regarded as having differential expression. The proteins identified were conducted in the Cluster of Orthologous Groups of proteins (KOG) analysis.

### 2.8. Bioinformatics Analysis

The gene ontology (GO) annotation analysis (http://www.geneontology.org/, accessed on 1 January 2023) was performed on the complete set of DEPs [30]. The Rattus norvegicus data was considered as the background, and the significance level was established as *p* < 0.05. To explore the potential biological pathways of the DEPs, the Kyoto Encyclopedia of Genes and Genomes (KEGG) database (http://www.genome.jp/kegg/, accessed on 1 January 2023) was utilized.

## 3. Results

### 3.1. Global Profiling of Proteins in Duodenum Tissue

As per the criteria for protein identification, a sum of 5594 proteins has been detected, each having one or more distinctive peptides with a *p*-value of less than 0.05. The number of proteins in which the peptides coverage ranging from 1% to 40% is about 4, 911 (Figure 1A). Of these, 36.8% (2059) of proteins were identified with one to two peptides, while 43.9% (2456) were identified with more than four unique peptides (Figure 1B). 65.92% (22,927) of peptides were in the 9–15 length range. The molecular weights of the proteins that were identified ranged from 1.347–846.683 kDa, in which 2804 proteins were about 21–60 kDa (Figure 1C).

### 3.2. KOG Annotation

The KOG database has ascertained the annotation of 5218 proteins that can be segregated into 25 distinct categories based on the functional classification assigned by the KOG. The proteins categorized as “general function prediction only” were the most prevalent, comprising 0.13% of the total and amounting to 672. There were 644 “signal transduction mechanisms”, 542 “posttranslational modifications protein turnover chaperones”, and 418 “intracellular trafficking, secretion, and vesicular transport”, accounting for 0.12%, 0.10%, and 0.08%, respectively (Figure 2).

### 3.3. Differential Expressed Proteins among Three Treatment Groups

Ratios of TMT reporter ion intensities were evaluated for common peptides from raw data sets to identify DEPs among boron treatment groups of 0, 10, and 320 mg/L. The computation of the FDR value for each *p*-value of DEPs was carried out along with the *t*-tests. Three groups, namely negative control (NC), B10, and B320, were referred to with different boron concentrations of 0, 10, and 320 mg/L, respectively. Among the comparison groups of NC vs. B10, NC vs. B320, and B10 vs. B320, 110, 32, and 179 proteins, respectively, demonstrated differential expression (Figure 3A–C, Appendix A). There were 74, 19, and 154 specific DEPs in NC vs. B10, NC vs. B320, and B10 vs. B320 comparison groups, respectively (Figure 3D). The expression pattern analysis revealed that out of the identified 284 DEPs, 254 proteins were predominantly clustered into two profiles among the ten profiles classified. Profiles 1 and 2 presented significant clustering patterns (Figure 4). Profiles 1 and 2 comprised 124 and 130 DEPs, respectively. The initial decrease in protein expression levels was observed in profile 1, followed by a slight increase as boron concentration increased. In contrast, profile 2 exhibited an initial increase in protein expression levels, which was followed by a slight decrease with increasing boron concentration.

### 3.4. GO Annotation of DEPs

The GO database is a universally recognized gene functional classification system utilized for providing a comprehensive depiction of the attributes pertaining to diverse genes and their associated products. A GO analysis was undertaken on all identified DEPs to gain improved comprehension. The proteins were categorized based on their participation in three significant classifications, namely biological process, cellular component, and molecular function, using second-level GO terms. In the NC vs. B10 comparison group, the DEPs were primarily associated with cell proliferation, cellular process, immune system process, metabolic process, multi-organism process, and response to stimulus in the biological process; the DEPs were mainly distributed in cell part, macromolecular complex, membrane, organelle, organelle part, and other organism part in cellular component; the DEPs were predominantly related to molecular function regulator and signal transducer activity in molecular function (Figure 5A, Appendix A).

In the NC vs. B320 comparison group, the DEPs were closely linked with cell proliferation, behavior, carbon utilization, locomotion, multi-organism process, and biological adhesion in biological process; the DEPs were mainly distributed in membrane, membrane part, and supramolecular complex in cellular component; the DEPs were mostly connected to signal transducer activity, molecular transducer activity, and transcription regulator activity in molecular function (Figure 5B, Appendix A).

In the B10 vs. B320 comparison group, the DEPs were predominantly linked to cell killing, cellular component organization or biogenesis, cellular process, developmental process, growth, immune system process, metabolic process, multicellular organismal process, multi-organism process, and response to stimulus in biological process; the DEPs were mainly distributed in cell part, extracellular region, extracellular region part, macromolecular complex, membrane-enclosed lumen, membrane part, organelle, and organelle part in cellular component; the DEPs were mostly associated with binding, structural molecule activity, transcription regulator activity, and transporter activity in molecular function (Figure 5C, Appendix A). More importantly, some proteins such as 2′-5′-oligoadenylate synthetase-like (OASL), myxovirus resistance 1 (MX1), myxovirus resistance 2 (MX2), dynein cytoplasmic 1 intermediate chain 1 (DYNC1I1), coiled-coil domain containing 88B (CCDC88B), bone marrow stromal cell antigen 2 (BST2), and immunoglobulin heavy constant mu (IGHM) were found to be involved in immune system process.

### 3.5. KEGG Enrichment Analysis of DEPs

After annotation and augmentation, a comprehensive analysis based on the KEGG pathway was conducted to identify potentially affected pathways due to differential protein expression in duodenum tissue, as various proteins work together to complete biological processes. In the NC vs. B10 comparison group, one hundred and ten DEPs were successfully mapped to a total of eighteen KEGG pathway entries. The analysis of the KEGG pathway revealed that the foremost ten pathways identified included herpes simplex infection, phagosome, autoimmune thyroid disease, graft-versus-host disease, allograft rejection, type I diabetes mellitus, antigen processing and presentation, human papillomavirus infection, cell adhesion molecules (CAMs), and necroptosis (Figure 6A, Appendix A).

In the NC vs. B320 comparison group, a total of 32 DEPs were mapped to four KEGG pathway entries, including phagosome, glycosaminoglycan biosynthesis, ribosome biogenesis in eukaryotes, and vitamin digestion and absorption (Figure 6B, Appendix A).

In the B10 vs. B320 comparison group, a total of 179 DEPs were mapped to 20 KEGG pathway entries. The analysis of the KEGG pathway revealed that the foremost ten pathways identified consisted of complement and coagulation cascades, systemic lupus erythematosus, staphylococcus aureus infection, protein digestion and absorption, pertussis, steroid hormone biosynthesis, renin-angiotensin system, antigen processing and presentation, prion diseases, and vitamin digestion and absorption (Figure 6C, Appendix A). Notably, some proteins, such as RT1 class Ib, locus S3 (RT1-S3), RT1 class Ia, locus A2 (RT1-A2), RT1 class I, locus CE7 (RT1-CE7), antigen peptide transporter 2 (LOC103689996), DYNC1I1, neutrophil cytosolic factor 4 (NCF4), Fc fragment of IgG receptor IIIa (FCGR3A), scavenger receptor class B, member 1 (SCARB1), signal transducer and activator of transcription 1 (STAT1) and signal transducer and activator of transcription 2 (STAT2), were mapped for pathways involved in the phagosome, antigen processing and presentation, and necroptosis pathway.

## 4. Discussion

Boron is a fundamental nutrient required for optimal growth and development of living organisms. Moreover, it has been established that boron assumes a significant role in the immune system’s response to various pathogens and exogenous stimuli [26]. Routray and Ali (2016) concluded that oral administration of boron at a physiological concentration (4.6 mg/kg) had a significant impact on enhancing the ability of the host to defend itself against infections and may also play a part in the prevention of cancer and other ailments in BALB/c mice [31]. Boron-containing compounds (4.6 mg/kg) administered to mice orally, such as boric and boronic acids, induced changes in lymphocytes and antibodies. The effects of the tested compounds on the lymphocyte populations and antibodies demonstrated variable outcomes, and there was a structure-activity relationship between these compounds as immunomodulatory drugs [32].

To conduct a more in-depth examination of the effects of boron on the intestine and to confirm the findings obtained from RNA-Seq data, protein profiling analysis of the duodenum was carried out using TMT peptide labeling quantitative proteomics based on previous transcriptome data. To differentiate the advantageous and detrimental effects of boron, boron-supplemented water was administered in low and high doses of 10 mg/L and 320 mg/L, respectively. Based on the threshold of *p*-value < 0.05, 5594 proteins were identified through proteome data, with most proteins having more than four unique peptides.

In the comparison groups of NC vs. B10, NC vs. B320, and B10 vs. B320, a total of 110, 32, and 179 DEPs were identified. The fold changes resulting from varying concentrations of boron in dietary supplementation fluctuated between 0.18 and 4.21. Notably, the proteins with the most significant fold difference were small nucleolar RNA 17 (AABR07060963.2), CCDC88B, OASL, and regulator of G-protein signaling 17 (RGS17). The expression pattern analysis unveiled that the majority of DEPs were predominantly clustered into two profiles. Furthermore, the expression trends of these proteins in the two profiles were consistent with the previous transcriptome outcomes [25]. The CCDC88B displays high levels of expression in various immune cells, such as CD4^+^ and CD8^+^ T lymphocytes and myeloid cells [33]. Its effects on various T lymphocyte functions encompass maturation, activation, division, and cytokine production in response to T cell receptor engagement [34]. CCDC88B possesses the capability to serve as a microtubule-associated protein and govern the migratory characteristics of lymphoid cells in intestinal inflammation by interacting with the dedicator of cytokinesis 8 (DOCK8) [35]. Further investigation is required to understand the specific mechanism behind the reduction in CCDC88B expression with the addition of 10 mg/L boron in the feed.

The duodenum proteome profiles identified among NC vs. B10, NC vs. B320, and B10 vs. B320 comparison groups mainly exhibited biological processes related to cell proliferation, cellular process, immune system process, metabolic process, multi-organism process, and response to stimulus. Notably, OASL, CCDC88B, MX1, MX2, and DYNC1I1 proteins were identified as being related to immune system processes. Sun et al. (2016) found that boron-supplemented water administered in low doses (80 mg/L) can promote the proliferation of intestine cells and inhibit apoptosis, while high doses of boron (320 and 640 mg/L) can increase apoptosis in the intestine and thus stimulate compensatory cell proliferation [36]. Xu et al. (2022) stated that drinking water with low and moderate boron concentrations (10–80 mg/L) enhances the microstructure, immunity, barrier function, antioxidant activity, and cell proliferation of the jejunum in rats, but high boron concentrations (480 and 640 mg/L) show negative effects [37]. These findings suggest that a suitable quantity of boron enhances intestinal immune function by stimulating cell proliferation, enhancing the microstructure of the barrier, and elevating antioxidant activity. These results offer a scientific basis for further investigation into the underlying mechanism.

Boric acid (0.1% (*w*/*w*)) supplemented in feed had the potential to regulate *Salmonella enteritidis* infection and thereby preserve intestinal homeostasis and microbiota balance in broiler chickens [38]. The establishment and survival of parasites could be adversely impacted by inadequate (0.2 mg/g) or minimal (2 mg/g) dietary boron consumption due to their influence on the microflora in the intestines [39]. Some recent studies have demonstrated that the autoinducer-2-borate (AI-2B), which is a signaling molecule that contains boron, plays a vital role in redressing the imbalances in microbiota composition through its impact on the intestinal flora [1,40,41]. The carbohydrate AI-2B is required by certain bacteria for communication purposes. Boron is also present in the colonic mucus gel layer of rats [42]. Naturally, organic boron species are considered potential prebiotic agents that can assist in the communication among bacteria through AI-2B and reinforce the mucus gel barrier within the human colon [1,42]. The development of dysbiosis and gut permeability may arise from the lack of boron during symbiosis, which results in a decline in AI-2B concentration [1,42]. Concomitantly, the reduction of boron levels in the colonic mucus promotes the generation of bacterial metabolites and pro-inflammatory cytokines, which may worsen the pathophysiology of osteoarthritis [43,44]. There exists the possibility for boron analysis derived from feces and colonic mucus to serve as a critical indicator of inadequate boron nutrition and a predictor for numerous diseases caused by an unhealthy symbiotic relationship in the future [1]. As a result, tracking the AI-2B marker in feces would introduce novel scientific prospects for disease prophylaxis concerning a dietary boron deficiency [1,45].

The 2′-5′ oligoadenylate synthase (OAS) and MX proteins, classified as interferon-stimulated genes, serve as effector molecules that facilitate innate immune responses against viral infections at the cellular level [46]. OAS and MX are widely recognized for their significant contribution to protecting the intestinal mucosa from viral infections [47,48]. The MX protein exhibits an extensive array of antiviral effects. The MX1 protein impedes the process of myxovirus replication, while the MX2 protein has a potent inhibitory impact on the vesicular stomatitis virus [49]. Recombinant IFN-β resulted in the expression of MX2 in human and mouse intestinal organoids [50]. As newborn piglets were infected with the porcine epidemic diarrhea virus, the expression of both antiviral genes MX1 and MX2 were highly significantly upregulated and peaked at 24 h post-infection, resulting in an intense inflammatory response [51]. The application of *Lactobacilli* on porcine intestinal epithelial cells resulted in a noteworthy elevation of OASL and MX2 levels [52,53]. The expression of OASL and MX2 genes was markedly elevated in 3D differentiated small intestinal and colonic organoid lines in comparison to their undifferentiated counterparts cultured in expansion media [54]. The B10 group exhibited a significant upregulation in the expression of OASL, MX1, and MX2 genes in comparison to the NC group, as per the current investigation. The proper incorporation of boron has the potential to enhance intestinal immunity, as evidenced by the study.

The gene pathway analysis using DEPs in the NC vs. B10 comparison group revealed activation of the phagosome, antigen processing and presentation, cell adhesion molecules (CAMs), and necroptosis. Zhang et al. (2021) demonstrated that the necroptosis pathway could significantly impact the immune response in boron-treated splenic lymphocytes of ostrich chicks (0, 0.01, 0.1, 0.5, 1, 5, 10, 25, 50, and 100 mmol/L) [55]. Conversely, DEPs in the NC vs. B320 comparison group showed the activation of the phagosome, glycosaminoglycan biosynthesis, ribosome biogenesis in eukaryotes, and vitamin digestion and absorption. Some DEPs were enriched in these pathways, including major histocompatibility complex (MHC) molecules, LOC103689996, FCGR3A, STAT1, STAT2, DYNC1I1, NCF4, and SCARB1.

The introduction of boron at a concentration of 10 mg/L led to an increase in antigen binding and presentation gene expression levels, activating the MHC-I pathway and regulating the function of CD8 T cells and natural killer cells, while the T cell receptor signaling pathway positively affected the intestinal environment’s preservation. The RT1 complex in rats plays a key role in peptide antigen binding, signaling receptor binding, and antigen processing and presentation through MHC class I and class II [56,57]. The B10 group displayed upregulation of RT1-A2, RT1-S3, and RT1-CE7 in contrast to the NC group. However, the MHC class I molecules, which include RT1-A1, RT1-A2, RT1-S3, RT1-CE7, and RT1-CE10, as well as the MHC class II molecules, such as RT1-Ba, RT1-Bb, RT1-Da, RT1-Db1, RT1-DMa, and RT1-DMb, did not exhibit any differences in the NC vs. B320 comparison group. The B320 group exhibited a downregulation of RT1-CE10 and RT1-Db1 in comparison to the NC group [58]. It has been suggested that the inclusion of boron in suitable quantities could potentially augment the immune response of the duodenum [24].

The LOC103689996 protein exhibited an upregulation in the B10 group when compared to the NC group. A study on glioblastoma tumors noted that the administration of 60 nm diameter gold nanoparticles (AuNPs) with Zpep led to an increase in its expression, ultimately benefiting smaller malignant gliomas [59]. FCGR3A, which functions as a receptor for the Fc region of immunoglobulin γ with low affinity, demonstrated upregulation in the NC vs. B10 comparison group [60]. Furthermore, it was identified as a predictive biomarker that was linked to tumor prognosis [60,61]. The interferon system contains the first two identified STAT proteins, which have a crucial function in regulating STAT-mediated signaling in host immune systems. Upregulation of STAT1 and STAT2 was observed in the NC vs. B10 comparison group, with STAT1 acting as a cytoplasmic transcription factor that could be activated by various stimuli and forming heterotrimers with STAT2 and interferon response factor 9 to induce expression of interferon-stimulated genes [62,63].

DYNC1I1 is responsible for the transport of protein-containing vesicles and organelles to the neuronal cell body through retrograde axonal transport [64]. Its deficiency could lead to neuronal atrophy in primary hippocampal neurons, and it has been found to be upregulated in hepatocellular carcinoma [65,66]. High expression of DYNC1I1 was associated with poor prognosis and promoted the proliferation and migration of gastric cancer cells by upregulating IL-6 expression, which suggests that DYNC1I1 may be a potential therapeutic target for gastric cancer [67,68]. This gene was enriched in the phagosome and downregulated in both B10 and B320 groups compared with the NC group. The expression of NCF4 was found to be reduced in the NC vs. B10 and NC vs. B320 comparison groups. In patients with Crohn’s disease who have rs4821544 variants in NCF4, there was a decline in reactive oxygen species following stimulation with the pro-inflammatory cytokine granulocyte-macrophage colony-stimulating factor [69]. The comparison between NC and B320 groups revealed an upregulation of SCARB1. Through analysis, a regulatory network was discovered that involved SCAT8/miR-125b-5p axis and contributed to the malignant progression of nasopharyngeal carcinoma by inducing SCARB1 expression [70]. Furthermore, SCARB1 was found to be linked to an increase in high-density lipoprotein-cholesterol and a heightened risk of myocardial infarction [71].

The administration of drinking water containing 320 mg/L elemental boron has been observed to cause toxicity to rats’ small intestines. Our hypothesis suggests that the organism improves metabolic efficiency to cope with high levels of elemental boron, allowing for survival in such an environment. It has been determined an increased level of boron is detrimental to cells, leading to the impairment of protein synthesis due to the stimulation of eIF2α phosphorylation in a GCN2 kinase-dependent manner [72]. An elevated concentration of a compound that contains boron has been observed to effectively hinder the accumulation of lipids in cells without any discernible toxic effects on said cells [73]. Further investigation is necessary to ascertain the optimal levels of supplementary boron intake that can effectively harmonize the digestive and absorptive capacity while eliciting an appropriate immune response.

## 5. Conclusions

In the present study, we demonstrated that appropriate doses of boron enhanced the immune response of the duodenum and interpreted the mechanism of physiological responses produced by different doses of boron on the duodenum from a proteomic point of view. We elucidated that a low concentration of 10 mg/L proved to be the most effective, whereas concentrations of 320 mg/L had the potential to suppress immune function. At the molecular level, the mechanism of the inhibitory effect of high-dose boron on immune function still needs to be further investigated, and further elucidation of the specific protein regulatory networks involved in different doses of boron will be helpful to reveal the nutritional and toxic effects of boron on animal organisms. Together, our findings provide new insights that the addition of appropriate doses of boron is beneficial to the growth and health of animals, providing theoretical references for the proper use of boron in human nutrition, as well as alleviating the pressure of over-addition of micronutrients on the environmental pollution of animal husbandry.

## Figures and Tables

**Figure 1 genes-14-01560-f001:**
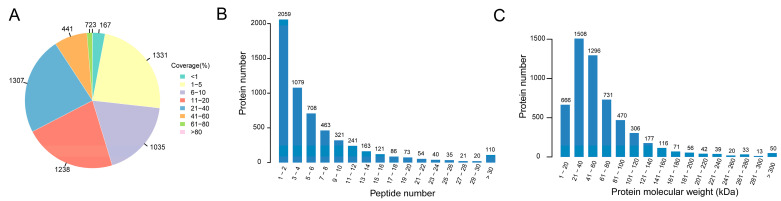
The characteristics of peptides identified in the sequenced data. (**A**) The distribution of coverage of the identified proteins. Each sector represents the proportion of a coverage range. The larger the sector, the greater the number of proteins in the coverage range, and the number outside the sector indicates the number of proteins in the coverage range. (**B**) The distribution of the number of peptides contained in the identified protein. The horizontal coordinates are the range of the number of peptides covering the protein, and the vertical coordinates are the number of proteins. (**C**) The molecular weight distribution of the identified proteins. The horizontal coordinates show the range of protein molecular weights, and the vertical coordinates show the number of proteins of the corresponding molecular weights.

**Figure 2 genes-14-01560-f002:**
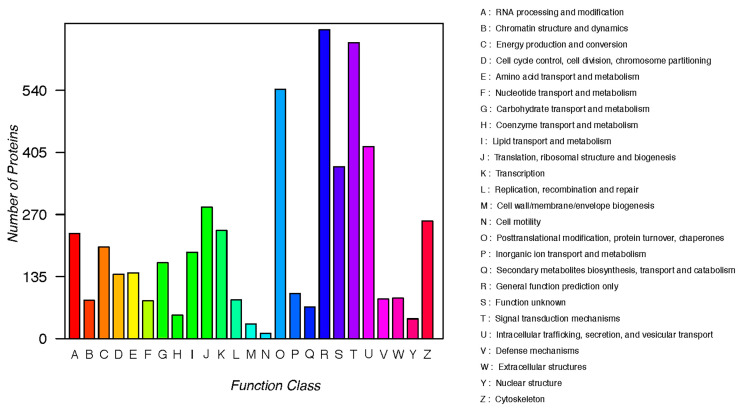
KOG function classification of all protein sequences. The horizontal coordinate represents the name of the KOG functional classification, and the vertical coordinate represents the number of proteins in the different classifications.

**Figure 3 genes-14-01560-f003:**
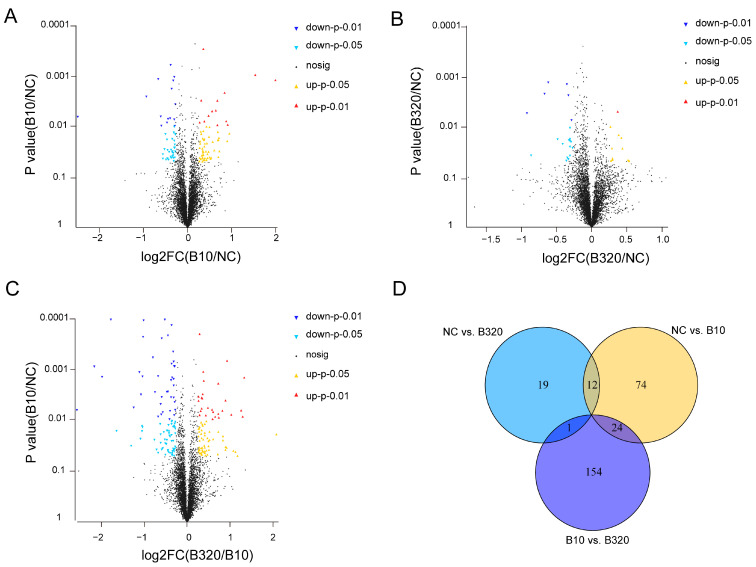
Profiles of DEPs after treatment with different concentrations of boron. Volcano plot of 110 DEPs between NC and B10 groups (**A**), 32 DEPs between NC and B320 groups (**B**), and 179 DEPs between B10 and B320 groups (**C**). Each dot in the graph represents a specific protein, yellow dots indicate proteins significantly upregulated at *p* < 0.05, red dots indicate proteins significantly upregulated at *p* < 0.01, wathet dots indicate proteins significantly downregulated at *p* < 0.05, mazarine dots indicate proteins significantly downregulated at *p* < 0.01, and black dots are non-significantly different proteins. (**D**) Venn plot of DEPs among NC, B10, and B320 groups.

**Figure 4 genes-14-01560-f004:**
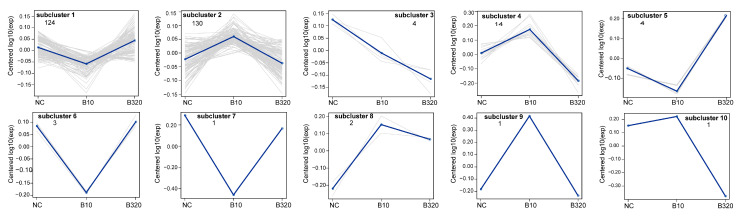
Sketch map of the cluster analysis of DEPs. The horizontal coordinates show each comparative sample group and the vertical coordinates show the expression level of the protein in this group of samples. Each line in the figure represents a protein, and the blue line represents the average expression level of all proteins in the sub-cluster. Each figure shows a type of expression pattern.

**Figure 5 genes-14-01560-f005:**
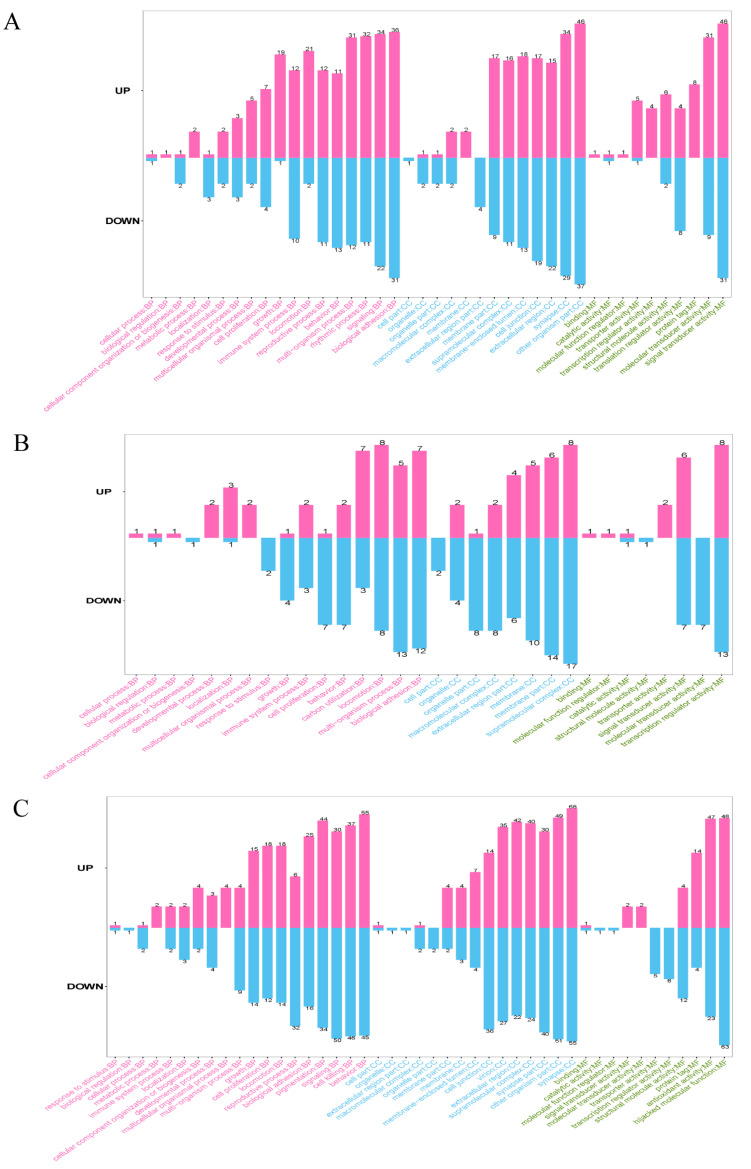
Annotation of DEPs using gene ontology after treatment with different concentrations of boron. Each bar in the diagram represents a secondary classification of GO. Red bars facing upward represent upregulated DEPs, and blue bars facing downward represent downregulated DEPs. The number on the bar represents the number of DEPs. The horizontal coordinates indicate the terms of the secondary classification of GO, with the capital letters in front of the terms being abbreviations for the three main categories (Biological Process (BP), Cellular Component (CC), and Molecular Function (MF)). The terms are labeled in different colors, with green for BP, blue for CC, and red for MF. (**A**) GO annotation of DEPs between NC and B10 groups. (**B**) GO annotation of DEPs between NC and B320 groups. (**C**) GO annotation of DEPs between B10 and B320 groups.

**Figure 6 genes-14-01560-f006:**
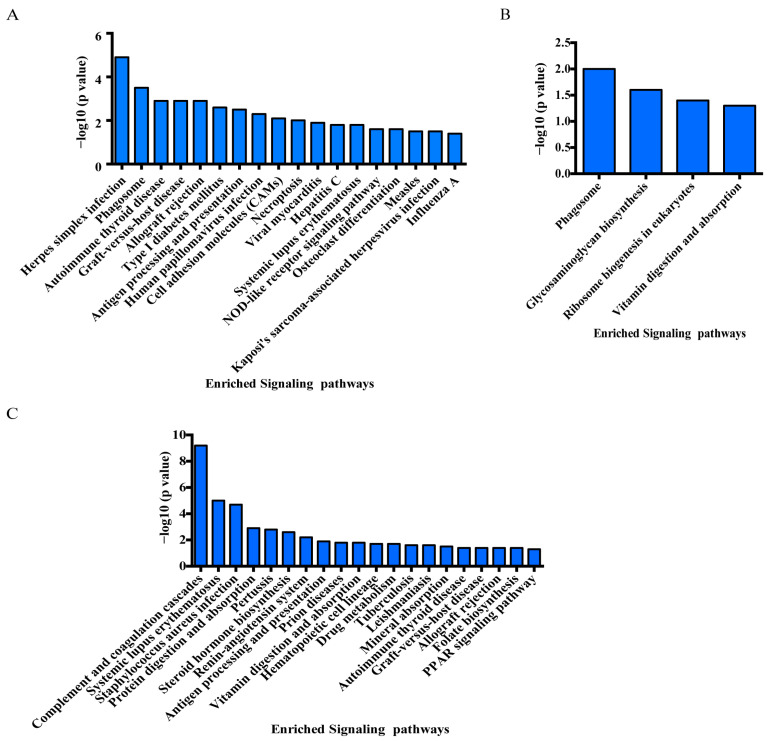
KEGG pathway analysis of DEPs after treatment with different concentrations of boron. The enriched signaling pathway of DEPs is exhibited in the lower axis and the proportion of DEPs (−log10 (*p*-value)) is listed in the upper axis. (**A**) Pathway analysis of DEPs between NC and B10 groups. (**B**) Pathway analysis of DEPs between NC and B320 groups. (**C**) Pathway analysis of DEPs between B10 and B320 groups.

## Data Availability

The datasets generated and/or analyzed during the current study are available from the corresponding author upon reasonable request.

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
