# Peer review of "Proteomic Analysis of Rat Duodenum Reveals the Modulatory Effect of Boron Supplementation on Immune Activity"

_genes, 2023, doi:10.3390/genes14081560_

Round 1

Reviewer 1 Report

The authors evaluated the method of TMT peptide labeling in conjunction with LC-MS/MS quantitative proteomics for the purpose of examining the effects of boric acid on a rat model, analyzing proteins from the duodenum. The results revealed that these differential expression proteins (DEPs) mainly clustered into two profiles. GO annotations suggested that most of the DEPs played a role in the immune system process, in which OASL, Mx1, Mx2, DYNC1I1, and CCDC88B showed differential expression. The DEPs had demonstrated an augmentation in the signaling pathways which primarily include phagosome, antigen processing and presentation, as well as cell adhesion molecules (CAMs).

The reviewer believes that this manuscript needs a very thorough revision.

In line 49 what does these abbreviations mean MOB1

In line 50 what does these abbreviations mean PD-L1

Section 1 "Introduction"

The reviewer believes that this section should be improved.

The introduction should be written in more detail, especially regarding the references on boron supplementation in the small intestine, specifically in the duodenum.

The importance of the duodenum and its anatomical, physiological, and immunological functions could be included.

The novelty of the study is not clear, so a clear and detailed explanation of how it differs from previous studies is needed.

At the end, add a paragraph about possible clinical and health applications.

Section 2. Methods

In sections 2.4. Trypsin digestion and TMT peptide labeling, 2.5. Reversed phase high-performance liquid chromatography (HPLC) analysis and 2.6. LC-MS/MS analysis. References should be added in the methodology. Or is this the first time that these experiments have been performed?

2.1. Experimental Animals and Design

Line 93 add the protocol number approved by the Anhui Laboratory Animal Care Committee

Lines 103-105 Why did they use these boron concentrations of 10 mg/L and 320 mg/L? Please add a reference to support these concentrations.

3. Results

The quality and presentation of all the figures is very poor. Please improve the quality and presentation the figures in all manuscript.

The Figures 1, 2, 5 and 6 could be improved or modified in other programs such as Sigma and Prisma.

The Figure 6 the axis of the ordinates is not detailed. Please add in the new version of graphics.

In lines 302 and 303 what does these abbreviations mean OASL, Mx1, Mx2, DYNC1I1, CCDC88B, BST2, and IGHM.

In lines 330 and 331 what does these abbreviations mean RT1-S3, RT1-A2, RT1-CE7, LOC103689996, DYNC1I1, NCF4, FCGR3A, SCARB1

3. Discussion

Results were good, while discussion is very short, and more details should be added as to discuss the whole obtained data.

In lines 353 and 354 The findings suggest that a suitable quantity of boron enhances intestinal immune function, providing a scientific basis for exploring the underlying mechanism”.  Could you explain this statement in more detail? Please check these references:

In line 390 the authors write “The coiled-coil domain containing protein 88b (CCDC88B)”, delete and add these sentences. The CCDC88B displays high……..

In line 399 the authors write “Dynein cytoplasmic 1 intermediate chain 1 (DYNC1I1)” delete and add these sentences: DYNC1I1 is responsible for the……..

In lines 421-423 the authors write “It has been suggested that the inclusion of boron in suitable quantities could potentially augment the immune response of the duodenum”. Could you explain this statement in more detail? Add references to support this.

A few issues would be best to address as potential variables:

1.Route of administration

2.Boron Concentration

5. Conclusion

The conclusion section appears to be a future direction. All conclusions must be convincing statements on what was found to be novel, impactful based on the strong support of the data/results/discussion. Moreover, the authors may be included the limitation of the present findings for a better understanding of the manuscript. And about possible clinical and health applications

Add section for abbreviations.

Minor editing of English lenguage required

Reviewer 2 Report

Observations:

       1. I would like to suggest that the following sentence to be rephrased, in order to ensure the correctness of reference literature:

“Boron is classified as a prebiotic micronutrient that is predominantly sourced from many kinds of food and drinking water in the human diet”

 A possible rephrase should be:

“Boron has been recently classified as a prebiotic micronutrient, being defined as essential in host–bacterial symbiosis in human health and is predominantly sourced from many kinds of food and drinking water in the human diet”

With the following references:

    2.  The following sentence should keep the essential literature reference, that also has to be added accordingly:

“Some recent studies have demonstrated that the autoinducer-2-borate (AI-2B), which

is a signaling molecule that contains boron, plays a vital role in redressing the imbalances

in microbiota composition through its impact on the intestinal flora [31,32]”.

   3. The authors are using literature, without doing a correct citation. For instance, the following sentence should quote the correct reference:

“Naturally organic boron species are considered as potential prebiotic agents that can assist in the communication among bacteria through AI-2B and reinforce the mucus gel barrier within the human colon. The development of dysbiosis and gut permeability may arise from the lack of boron during symbiosis, which results in a decline in AI-2B concentration”.

So the correct references are:

i) New insights into boron essentiality in humans and animals. Int. J. Mol. Sci. 2022, 23, 9147];

ii) “Diester Chlorogenoborate Complex: A New Naturally Occurring Boron-Containing Compound. Inorganics 2023, 11, 112].

   4. The authors keep using arguments from literature, without a correct citation:

“There exists the possibility for boron analysis derived from feces and colonic mucus to serve as a critical indicator of inadequate boron nutrition and a predictor for numerous diseases caused by an unhealthy symbiotic relationship in the future [36]. As a result, tracking the AI-2B marker in feces would introduce novel scientific prospects for disease prophylaxis concerning a dietary boron deficiency [37]”.

The correct citation after the above sentence is the following:

i)                https://www.mdpi.com/1422-0067/23/16/9147;

ii)              https://patents.google.com/patent/WO2023070074A1/en .

Subsequently, the existing references need to be removed:   

36. Vijay, A.; Valdes, A.M. Role of the gut microbiome in chronic diseases: a narrative review. European Journal of Clinical Nutrition 2022, 76, 489-501, doi:10.1038/s41430-021-00991-6.

37. Durack, J.A.-O.; Lynch, S.V. The gut microbiome: Relationships with disease and opportunities for therapy.

Round 2

Reviewer 1 Report

I agree with the authors' responses and my recommendation is that it be accepted in its current form.

N/A